# Can a Crop Rotation and Fallow System Reduce the Carbon Emission Intensity of Agriculture?

Xuefeng Zhang [1,2], Hui Sun [1,2,*], Xuechao Xia [1,2], Zedong Yang [1,2] and Shusen Zhu [1,2]

1 Xinjiang Innovation Management Research Center, Xinjiang University, Urumqi 830046, China; 107552200191@stu.xju.edu.cn (X.Z.); xiaxuechao99@stu.xju.edu.cn (X.X.); ambitiousyzd@stu.xju.edu.cn (Z.Y.); zhushusen@stu.xju.edu.cn (S.Z.)
2 School of Economics and Management, Xinjiang University, Urumqi 830046, China
* Correspondence: shui@xju.edu.cn

**Abstract:** Under the carbon emission pattern of "peak carbon and carbon neutrality", the policy of crop rotation and fallow system (CRFS) is regarded as an important initiative to promote the green, low-carbon, and high-quality development of agriculture. Focusing on balanced panel data from 30 provinces in China from 2010 to 2021, this paper empirically examines the impact of CRFS on agricultural carbon emissions (ACEI) and its internal mechanism using a multi-temporal difference-in-differences model. The benchmark regression results show that CRFS can significantly reduce ACEI, and the results remain robust after validation by multiple methods. Mechanism results show that CRFS is able to reduce ACEI by reducing factor mismatch and promoting the level of agricultural services. Heterogeneity analysis results show that the arable land fallow rotation system is more conducive to promoting the reduction in agricultural carbon emission intensity in the main grain producing areas, main grain marketing areas, high land transfer areas, and plantation areas than in the grain production and marketing balanced areas, low land transfer, and animal husbandry areas. This study demonstrates the effectiveness of the CRFS policy implementation, provides a doctrinal basis for expanding the scope of CRFS implementation, and provides policy recommendations for relevant departments to improve the CRFS policy.

**Keywords:** policy assessment; multi-phase did model; resource allocation; convergence analysis





## 1. Introduction

The Implementation Plan for Emission Reduction and Carbon Sequestration in Agriculture and Rural Areas issued in July 2022 clearly states that accelerating the modernisation of agriculture and rural areas should be the leading role, taking the green and low-carbon development of agriculture and rural areas, the implementation of major actions on pollution reduction and carbon reduction, and the enhancement of carbon sinks as the key and setting up a perfect monitoring and evaluation system. The 20th report states the following: "Strengthen the foundation of food security on all fronts, and ensure that the rice bowls of the Chinese people are firmly in their own hands". While guaranteeing the growth of food production, how to achieve the goal of "double carbon" and reach the win-win situation of food security, carbon reduction, and pollution reduction is a question that needs to be answered urgently for the current agricultural production. Therefore, it is of great significance for China to achieve the synergistic development of the two through optimising land cultivation patterns. However, China faces a series of difficulties in arable land protection, with increasing climate instability, declining arable land quality, serious soil erosion, and weakened land ecosystem restoration capacity, which seriously hinder the sustainable development of Chinese agriculture. In order to ensure the safety of arable land and promote the sustainable development of agriculture, and to reduce agricultural carbon emission intensity (ACEI), the Chinese government has continuously adjusted land use patterns and improved agricultural development policies [1]. Since 2016, the Chinese

government has launched a pilot crop rotation and fallow system (CRFS) in some provinces. In 2018, the Chinese government further expanded the scope of the CRFS pilot programme. In order to fully realise the effect of CRFS, the Chinese government has included CRFS in the 14th Five-Year Plan and asked governments at all levels to improve the supporting measures of CRFS in order to promote the sustainable development of agriculture. So, can CRFS improve ACEI?

It is a measure of changing crops (crop rotation) or not ploughing (fallow) for a certain period of time for the purpose of the protection, nourishment, and restoration of soil strength in order to improve the efficiency of farming and achieve the sustainable use of arable land [2]. However, the impact of CRFS on the carbon intensity of agriculture is uncertain. On the one hand, from the perspective of the general pattern of production activities, fallow will reduce the operation of agricultural production activities and reduce the input of agricultural production factors [3], which may lead to a decrease in ACEI. On the other hand, crop rotation changes the order and frequency of regular agricultural cultivation, increasing the instability of crop cultivation activities, and it may also increase agricultural carbon emissions due to the fuel consumption generated in the process of agricultural mechanisation, which may in turn increase ACEI. Based on the above analysis, it is difficult to make a clear judgement from the theoretical level on the carbon emission reduction effect of the implementation of the CRFS policy, so the answer to this question needs to be further explored empirically. Since the pilot work of CRFS started late in China, some rural areas lack an understanding of the effect of CRFS. Due to the fear that CRFS will lead to a decrease in crop yields and farm income, part of the farming population resists the promotion of CRFS to a certain extent. This not only seriously hinders the improvement of arable land quality but also affects the sustainable development of agriculture. Therefore, it is of great practical significance to reveal the impact of CRFS on CRFS and its internal mechanism, which is not only conducive to understanding the policy effects of CRFS in China, but also to facilitate the promotion of CRFS.

The aim is to increase the sustainable productivity of arable land. Fallowing has an impact on microbial population enhancement, water conservation, ecosystem restoration, and agricultural labour recuperation. Fallowing not only reduces the risk to soil biomes from chemicals such as pesticides and plastics but also promotes the growth of beneficial mycorrhizal fungi through soil mulching, which in turn promotes land ecosystem resilience [4,5]. Water resources are an important factor affecting the use of arable land. Fallowing gives a period of reprieve in the natural supply system of water resources and a rise in the water table of soil groundwater resources [6]. Fallowing avoids the over-consumption of agricultural labour and promotes the recovery of the physical quality of the labour force. The purpose of crop rotation is to regulate the physicochemical properties of soil and improve soil ecology to maintain soil fertility [7]. Crop rotation can not only alleviate the over-absorption of chemical elements but also increase soil chemical elements through the biochemical function of plants, so as to achieve calm soil physicochemical properties [8] and thus curb the intensity of agricultural carbon emissions.

China opened the CRFS pilot in 2016 and expanded it in 2018 and 2021. This provides a high-quality sample of natural experiments to explore the impact of CRFS on ACEI. The possible contributions of this paper are mainly as follows: compared with existing studies, this paper deepens the following three aspects. First, based on the theory of economies of scale and the theory of the substitution effect, this study deeply explores the potential mechanism of the CRFS policy affecting agricultural carbon emissions from the aspects of the level of resource factor mismatch and agricultural socialised services. This helps to comprehensively and deeply explore how the CRFS policy affects agricultural carbon emissions, with a view of enriching related research. Secondly, this paper takes into account that there may be differences in the impacts of CRFS policy on agricultural carbon emissions in areas with a different positioning of agricultural functional zones, different degrees of land transfer, and different precipitation, and it further analyses the heterogeneity of the policy effects under different degrees of land transfer and the positioning of agricultural

functional zones, which provides a direction for research on how CRFS reduces the intensity of agricultural carbon emissions. Thirdly, this paper analyses the important impact of the CRFS policy on regional agricultural carbon emissions from the perspective of convergence, $\beta$. The absolute convergence test results show that there is a significant club convergence feature in the level of agricultural carbon emissions, and the conditional $\beta$ convergence results show that the CRFS policy not only reduces the agricultural carbon emissions, but also promotes the convergence of the level of its regional carbon emissions.

The aim of this paper is to explore in depth the effects and mechanisms of crop rotation and fallow systems on agricultural carbon emissions, so as to provide references for optimal policy formulation and agricultural carbon emission reduction. The structure of the paper is arranged as follows: the second part covers the theoretical analysis and research hypotheses; the third part covers the model setting and variable selection; the fourth part covers the results; the fifth part is the discussion section; the sixth part is the conclusion of the paper; and the last part covers the policy recommendations.

## 2. Theoretical Analysis and Research Hypothesis

### 2.1. Policy Background and Literature Review

The existing literature mainly argues that the policy effects of CRFS can be summarised as ecological benefits, environmental benefits, and economic benefits. (1) Ecological benefits. Most studies have found that CRFS promotes the stability of the microbial population structure and abundance by maintaining the soil surface temperature and water content in the soil as well as the necessary conditions for the survival of fungi through soil surface mulch [9]. (2) Environmental benefits. On the one hand, traditional cultivation requires large quantities of pesticides, fertilisers, and chemicals such as plastic films. This not only emits pollutants during use but also destroys the restorative power of the land ecosystem. Reducing the use of chemical products promotes the self-repairing power of the land ecosystem, thus reducing the use of chemical products and pollution emissions. On the other hand, crop rotation can promote plants to fix chemical elements such as nitrogen, phosphorus, and potassium in the atmosphere, compact the soil, and thus reduce pollution emissions, which makes the carbon emissions at the agricultural level reduced [10]. (3) Economic benefits. By maintaining soil water content, nitrogen, phosphorus, potassium, and other components necessary for crop growth, it can promote the restoration of soil strength and then promote crop yield improvement to obtain economic benefits [11,12]. Crop rotation reduces external shocks by promoting internal nutrient cycling in arable land, maintaining the long-term productivity of arable land and breaking the disease cycle, enhancing the carbon adsorption capacity of arable land and improving land resilience.

Through the fallow system, the natural environment of the United States has been greatly improved, and the soil utility has been significantly enhanced. Canada, on the one hand, alleviates the pressure of a food surplus, and on the other hand, the straw return to the next year's fallow mode ensures that the soil fertility can be restored [13]. Japan's production mode has been transformed, and flowers and vegetables have begun to be planted in the farmland [14]. The implementation of China's policy can help protect arable land resources, promote the transformation of traditional agriculture to modern agriculture, reasonably adjust the structure and proportion of China's planting industry, and achieve the coordinated development of China's agricultural regions. However, the implementation of fallow policy has also attracted criticism [15]. On the one hand, fallow will objectively lead to a reduction in food production, and it is necessary to control the scale of fallow in total. On the other hand, fallow farming will cause certain economic losses to the contracted operators of arable land, and if the government intervenes too much in the administration of crop rotation and fallow farming, the result will be low economic efficiency. In addition, the adoption of a high subsidy policy for agricultural fallow will also lead to a large amount of land resources being left idle [16]. The optimisation of the cost-effectiveness of crop rotation and fallow can only be achieved through the effective combination of policy and market.

Scholars have conducted relevant studies on agricultural carbon emissions and their intensity, which are mainly divided into the following three aspects. First, most scholars measure agricultural carbon emissions and their intensity [17–19] and at the same time analyse regional differences and dynamic evolution patterns on the basis of measurement [20,21]. Second, the impact path is considered. Most studies find that the structural adjustment of the agricultural industry [22,23], population size [24], agricultural technology innovation [25], and agricultural production efficiency [26] are the main paths to achieve carbon emission reduction. Third, the national policy system is considered. With the depth of research, more and more scholars have found that agricultural production subsidies [27], land transfer policies [28], policies for major grain producing areas [29], and land leasing systems [30] are important influencing factors for ACEI.

Existing research on the impact of CRFS on the level of carbon emissions from agriculture has also been explored, with most of the literature demonstrating the impact of fallow crop rotation on the resource efficiency of input factors. Some scholars have found that fallow crop rotation not only promotes plant nitrogen fixation and reduces the use of chemical fertilisers but also reduces the volatilisation of nitrogen fertilisers due to plant nitrogen fixation, thus improving the efficiency of nitrogen fertiliser use [31,32]. Water is a necessary resource for crop growth and an important factor affecting crop yield. Research has pointed out that fallow crop rotation can promote the water storage capacity of soil, ensure the water level of groundwater resources, and thus promote the efficiency of water resources [33]. As the growth carrier of crops, land resources have always been the focus of scholars. Some scholars have found through experiments that the soil mulch produced due to fallow cultivation can promote the ecosystem restoration function of soil mulch, promote the increase in biological populations, enrich the regional biodiversity, and thus inhibit the level of agricultural carbon emissions [34].

In summary, there are some shortcomings in the research on CRFS and ACEI: First, the existing literature does not directly answer the what the effect of CRFS is on ACEI. The implementation of CRFS in China is late, and the effect of agricultural carbon intensity has to be assessed. This also puts forward the demand for subject validation for academics to argue the effect of CRFS policy. Secondly, the existing research focuses on analysing the internal mechanism of the benefits of the CRFS policy from a theoretical perspective, and some scholars have analysed the impact of CRFS through experiments, while few studies have examined the impact of CRFS on CRFS and its internal mechanism from a qualitative empirical perspective.

### 2.2. Research Hypothesis

For a long time, under the pressure of the supply of agricultural products, there has been an excessive depletion of arable land, over-exploitation of groundwater, and heavy use of chemical fertilisers and pesticides, as well as agricultural machinery, which in turn has led to increased carbon emissions from the agricultural production process. The implementation of CRFS can affect ACEI by changing soil fertility, resource allocation, and soil remediation mechanisms. First, soil fertility conservation mechanisms are considered. Rotational ploughing can achieve the rotation of cereal crops, legume crops, dryland crops, paddy crops, etc., which can regulate soil physicochemical properties and improve soil ecological conditions [35]. The implementation of corn and soybean crop rotation can bring into play the roles of soybean rhizoma nitrogen fixation and land cultivation and fertilisation, thus realising the combination of planting and land cultivation. Fallowing allows arable land to restore the soil strength through natural ecosystems, promotes the rise of the water table, and improves the strength of arable land through the combination of land nourishment and cultivation [36]. Second, resource allocation mechanisms are considered. There is an urgent need to promote the structural reform of the agricultural supply side, through the arable land fallow crop rotation; the saving and efficient use of resources; adjusting and optimising the planting structure; increasing the supply of scarce agricultural products; meeting diversified consumer demand; and comprehensively improving the

quality and efficiency of the agricultural supply system. Third, soil restoration mechanisms are considered. The following are the aims of arable land fallow crop rotation: reduce the intensity of exploitation and utilisation; reduce chemical fertiliser and pesticide inputs; alleviate the pressure on the ecological environment; and facilitate soil restoration, so as to inhibit the increase in the intensity of agricultural carbon emissions [37]. Based on the above analysis, this paper puts forward the following hypothesis:

**H1:** *The implementation of CRFS has favoured the reduction in ACEI.*

Inefficient agricultural resource use and production practices are a direct cause of high carbon emissions in agriculture [38]. The implementation of CRFS can reduce the mismatch of agricultural resources affecting ACEI, which is reflected in chemical inputs, water inputs, and labour inputs. Firstly, traditionally, large amounts of pesticides are sown to address crop pests and weeds, and large amounts of fertilisers are sown to provide chemical elements for crop growth. However, this directly destroys the original physicochemical properties and biological population structure of arable land, which in turn reduces the sustainable use of land; fallow reduces the amount of pesticides and chemical fertilisers and other chemical inputs, avoids the continuous destruction of the soil's physicochemical properties and structure, and improves the ability of the land to sequester carbon. Fallowing encourages soil cover to play a protective role and repair the natural soil fertility [39]. Crop rotation avoids the over-absorption of a certain chemical element by a certain crop and brings into play the nitrogen-fixing function of plants, and at the same time, crop rotation also reduces the transmission of plant diseases and pests and thus reduces the use of pesticides. Therefore, CRFS reduces the input of chemicals and reduces carbon emissions from the source of land. Secondly, providing water for crops consumes a large amount of water resources but also leads to a serious depletion of freshwater resources in China. Reducing the over-absorption of water by crops promotes a rise in the surface water table, which increases soil wetness and reduces the need for crop irrigation [40]. Thirdly, prolonged labour increases the morbidity rate of the agricultural workforce and even reduces the life expectancy of the workforce, thus lowering the quality and supply of labour. Fallowing allows the working population a period of recuperation and facilitates the recovery of the physical quality of the population. This optimises the supply of agricultural workers in terms of individual quality and overall quantity, reduces the mismatch of agricultural labour resources, changes the inertia pattern of agricultural labour use [41], forms the endogenous impetus to promote the low-carbon transformation of agriculture, and corrects the level of resource mismatch at the level of labour, thus reducing the level of agricultural carbon emissions. Based on the above analysis, this paper puts forward the following hypothesis:

**H2:** *CRFS reduces agricultural carbon emission intensity by correcting the degree of resource mismatch.*

In order to promote the implementation of the CRFS policy, the government has provided substantial financial support for agricultural socialisation services, which represents the government's concern for agricultural development and the direction of agricultural development, which in turn influences the use of various services in the agricultural production process, with a view of promoting the transformation of agricultural modernisation and achieving the "dual-carbon goal". The implementation of the policy can enhance the level of agricultural socialised services in terms of the substitution effect of agricultural financial subsidies and the scale effect of the land management scale, and it further reduces the level of agricultural carbon emissions. On the one hand, studies have shown that agricultural socialised services can reduce the incentive for farmers to blindly purchase and use environmentally damaging factors of production through the substitution effect of reduced government intervention in the form of financial subsidies, thereby reducing agricultural carbon emissions [42]. On the other hand, agricultural socialisation services can reduce the intensity of agricultural carbon emissions by reasonably adjusting the factor input level and optimising the agricultural planting structure, so as to form a scale effect

of the land operation scale, operate the land in a continuous area, intensively utilise the factors of production, and alleviate the negative factor production efficiency. Based on the above analysis, this paper puts forward the following hypothesis:

**H3:** *CRFS reduces agricultural carbon intensity by improving agricultural socialised services.*

## 3. Model Setting and Variable Selection

### 3.1. Model Setting

3.1.1. Base Regression Model

In the Traditional and Classic DID model settings, all individuals in the treatment group are subjected to the same time of policy shock; but in fact, many policies are implemented in different regions and at different times. The CRFS policy began to be regulated and implemented nationwide in 2016. Due to the different soil and water resource conditions and agricultural industry development needs of each province (autonomous regions and municipalities directly under the central government), the progress of the implementation of the CRFS policy varies greatly from province to province, and the time of implementation of the policy varies from province to province. In view of the incomplete consistency of individual treatment periods in the study of the CRFS policy, this paper constructs the following multi-period DID model under the estimation framework of double fixed effects in order to identify the impacts of the implementation of the CRFS policy on agricultural carbon emissions:

$$ACEI_{it} = \alpha_0 + \beta_0 treat \times post_{it} + \chi_0 \sum_{i=1} Xit + \eta_{it} + \kappa_{it} + \varepsilon_{it} \qquad (1)$$

where $i$ denotes the province; $t$ denotes the period; $ACEI_{it}$ denotes the intensity of agricultural carbon emissions in the province $i$ in the period $t$; $treat \times post_{it}$ denotes a dummy variable for the treatment period that varies by individual; if an individual $i$ receives the treatment in period $t$, which represents its entry into the treatment period, then it takes the value of 0 in all previous periods and the value of 1 in all periods thereafter, with the coefficient $\beta_0$ being the average treatment effect for the whole group to which attention should be paid, and $\sum_{i=1} X_{it}$ denotes the control variable; $\eta_{it}$ denotes the province fixed effect; $\kappa_{it}$ denotes the year fixed effect and is a constant term, and the effects capture inter-individual differences that do not vary over time as well as the problem of omitted variables that do not vary with individuals but do vary over time; $\alpha_0$ denotes the random error term; $\alpha_0$ is the constant term; and $\beta_0$ and $\chi_0$ are the parameters to be estimated. Equation (1) controls for two-way fixed effects, and the estimated parameter $\beta_0$ is the net effect of the implementation of the CRFS policy on agricultural carbon emissions.

3.1.2. Event Study Method

The multiplicative difference method was used to examine the impact of the CRFS policy on the intensity of agricultural carbon emissions on the premise that there should be no significant trend difference between the intensity of agricultural carbon emissions in pilot provinces and non-pilot provinces before the implementation of the policy. Therefore, the study adopts the event study method to examine the trend of its change, and the model setup is as follows:

$$ACEI_{it} = \alpha + \sum_{t=-6}^{t=4+} \beta_t * D_{it} + \rho X_{it} + u_t + \delta_i + \varepsilon_{it} \qquad (2)$$

where $D_{it}$ is a series of dummy variables indicating the policy variables before and after the relative pilot year, where the subscripts take negative values to indicate the time before the start of the pilot of the CRFS policy and take zero and positive values to indicate the year of the pilot implementation and the subsequent years.

### 3.1.3. Mechanism Testing Models

In this paper, the mediated effects model is used to verify the intrinsic mechanism by which the CRFS policy affects the intensity of agricultural carbon emissions. The first stage verifies the impact of CRFS policy on agricultural resource mismatch coupling synergy and agricultural socialisation services. The second stage verifies the impact of agricultural resource mismatch coupling synergy and agricultural socialisation services on agricultural carbon emission intensity. The mechanism validation model of this paper is set as follows:

$$Med_{it} = \alpha_0 + \beta_0 treat \times post_{it} + \chi_0 Contr_{it} + \eta_{it} + \kappa_{it} + \varepsilon_{it} \tag{3}$$

$$ACEI_{it} = \alpha_0 + \beta_0 Med_{it} + \chi_0 Contr_{it} + \eta_{it} + \kappa_{it} + \varepsilon_{it} \tag{4}$$

where $Med_{it}$ denotes the mechanism variables, including the agricultural resource mismatch coupling synergy index and agricultural socialisation services; $\beta_0$ is the parameter to be estimated; and the rest of the variables and coefficients are set in the same way as in Equation (1).

### 3.2. Variable Selection and Data Description

#### 3.2.1. Explanatory Variable: Agricultural Carbon Intensity (ACEI)

In this paper, agriculture in the narrow sense (i.e., plantation) is taken as the object of study and its agricultural carbon emissions are measured. Referring to the study of Yadav et al. (2018) [43], it is considered that the agricultural carbon sources mainly include six categories of fertilisers, pesticides, agricultural films, agricultural diesel, agricultural land crop area, and agricultural irrigated area, and their carbon emission coefficients are $0.8956 \text{ kg} \cdot \text{kg}^{-1}$, $4.9341 \text{ kg} \cdot \text{kg}^{-1}$, $5.1800 \text{ kg} \cdot \text{kg}^{-1}$, $0.5927 \text{ kg} \cdot \text{kg}^{-1}$, $3.1260 \text{ kg} \cdot \text{hm}^{-2}$, and $26.6500 \text{ kg} \cdot \text{hm}^{-2}$. The carbon intensity of agriculture is measured by the ratio of agricultural carbon emissions to the total agricultural output value. Carbon intensity reflects the carbon emissions per unit of output value, which is a better measure of the coordination between economic development and environmental protection than the total carbon emissions [44].

#### 3.2.2. Core Explanatory Variable: Cropland Fallow Rotation System (*CRFS*)

In this paper, the proxy variables of the pilot policy of the CRFS are measured by multiplying the dummy variables of the pilot provinces and the time dummy variables before and after the implementation of the policy. The pilot policy of the arable land fallow and rotational cropping system has been implemented several times, in order to ensure the scientificity of the selection of experimental groups, and this paper takes the provinces of the pilot policy of the arable land fallow and rotational cropping system with multiple batches as the "treatment group". The time of implementation of the pilot policy of CRFS is 2016, 2017, and 2019,. For the non-CRFS pilot policy samples, $treat \times post_{it}$ is set to zero.

#### 3.2.3. Control Variables

This paper synthesises the research on the factors affecting agricultural carbon emissions mentioned in the previous section and controls for important variables affecting the intensity of agricultural carbon emissions. These include the following: (1) planting efficiency (PE) is the grain output/total area sown with grain, which measures the cultivation efficiency induced by land fertility; (2) the soil erosion control level (SEC) is the natural logarithm of the erosion control area +1, which measures the ability of rural areas to cope with natural disasters; (3) agricultural policies (BF) use agricultural subsidies/rural employees to measure the unit level of agricultural support; (4) the size of employment (SE) uses the number of rural employees/rural population to measure the level of rural labour force employment; (5) the degree of disaster (ED) is the affected area/total sown area of crops, which measures the occurrence of disasters in the agricultural production process; (6) the agricultural output per capita (AOP) is the total agricultural output/rural population, which characterises the level of rural economic development. The description and descriptive statistics of each variable are shown in Table 1.

**Table 1.** Description and descriptive statistics of variables related to the impact of CRFS policy on agricultural carbon emissions.

| | Variable Symbol | Description of Variables | Observed Value | Mean | Std. Dev. | Min | Max |
|---|---|---|---|---|---|---|---|
| Explanatory variable | $ACEI$ | Carbon intensity of agriculture | 360 | 5.47 | 1.02 | 2.73 | 6.90 |
| | $ACE$ | Agricultural carbon emissions | 360 | 338.81 | 230.04 | 15.35 | 996.75 |
| Explanatory variable | $treat \times post$ | Policy interaction term | 360 | 0.37 | 0.48 | 0 | 1 |
| | $PE$ | Planting efficiency | 360 | 0.36 | 0.10 | 0.18 | 0.69 |
| | $ED$ | Degree of disaster | 360 | 0.15 | 0.14 | −0.45 | 0.78 |
| | $AOP$ | Per capita agricultural output | 360 | 1.01 | 0.54 | 0.26 | 3.82 |
| Control variable | $BF$ | Policy on favouring agriculture | 360 | 0.51 | 0.70 | 0.11 | 5.71 |
| | $SE$ | Scale of practice | 360 | 0.017 | 0.02 | 0.003 | 0.07 |
| | $SEC$ | Soil erosion control | 360 | 7.64 | 1.86 | 0 | 9.67 |
| | $CP$ | Resource mismatch coupling synergy | 360 | 0.87 | 0.17 | 0.12 | 1 |
| Mechanism variables | $Service$ | Agricultural socialisation services | 360 | 0.81 | 0.56 | 0.18 | 3.18 |

### 3.2.4. Mechanism Variables

(1) Agricultural resource mismatch. The resource mismatch index mainly involves agricultural capital resource mismatch and agricultural labour mismatch. Referring to Wang et al. (2023) [45], the capital mismatch index and human capital index mismatch are as follows:

$$\tau_{Kit} = 1/\gamma_{Kit} - 1, \ \tau_{Hit} = 1/\gamma_{Hit} - 1 \tag{5}$$

where $\gamma_{Kit}$ and $\gamma_{Hit}$ are indices of absolute factor price distortions, which are replaced by coefficients of relative price distortions because they are not observable in practice:

$$\hat{\gamma}_{Kit} = \left(\frac{K_{it}}{K_t}\right) / \left(\frac{s_{it}\beta_{Kit}}{\beta_{Kt}}\right), \ \hat{\gamma}_{Hit} = \left(\frac{H_{it}}{H_t}\right) / \left(\frac{s_{it}\beta_{Hit}}{\beta_{Ht}}\right) \tag{6}$$

where $K_{it}/K_t$ is the share of physical capital actually used in the $i$ region, $ts_{it} = p_{it}y_{it}/Y_i$ is the share of output in the $i$ region in $t$, $\beta_{Kt} = \sum s_i\beta_{Kt}$ is the value of the capital contribution of formaldehyde to output in $t$, and $s_{it}\beta_{Kit}/\beta_{Kt}$ is the share of capital used in revenue in the $i$ region in $t$ in the case of an efficient allocation of physical capital. The ratio of the two is the degree of deviation between the actual share of physical capital used in $i$ and the effective allocation of physical capital in $t$, i.e., the degree of physical capital mismatch. The corresponding indicator for human capital has the same meaning. The $\beta_{Kit}$ and $\beta_{Hit}$ are further estimated using a CD function with constant returns to scale:

$$Y_{it} = AK_{it}^{\beta_{Ki}}H_{it}^{1-\beta_{Hi}} \tag{7}$$

Collate and add fixed effects to obtain:

$$\ln(Y_{it}/H_{it}) = \ln A + \beta_{Ki}\ln(K_{it}/H_{it}) + \mu_i + \lambda_t + \varepsilon_{it} \tag{8}$$

The aggregate output ($Y_{it}$) uses real GDP. Human capital input ($H_{it}$) uses the product of average annual employment and average years of schooling. Physical capital inputs ($K_{it}$) use the fixed capital stock:

$$K_t = I_t/P_t + (1-\delta)_t K_{t-1} \tag{9}$$

$I_t$ is the nominal fixed asset formation, $P$ is the fixed asset investment value index, $\delta_t$ is the depreciation rate, which is taken as 9.6 percent using the LSDV method to estimate Equations (4), (5) and (7) are brought in to obtain $\tau$. $\tau > 0$ represents the under-allocation of resources and vice versa for over-allocation.

This paper further uses a coupled synergy model to measure the composite index of resource mismatch as follows:

$$C_{it} = \frac{2\sqrt{\tau_{Kit} \times \tau_{Hit}}}{\tau_{Kit} + \tau_{Hit}} \tag{10}$$

$$T_{it} = 0.5 \times \tau_{Kit} + 0.5 \times \tau_{Hit} \tag{11}$$

$$D_{it} = \sqrt{T_{it} \times C_{it}} \tag{12}$$

where $C_{it}$ is the degree of coordination, $T_{it}$ is the degree of coupling, and $D_{it}$ is the degree of coupling coordination, which represents the combined value of physical capital mismatch and human capital mismatch.

(2) Agricultural socialisation services. Measured by the ratio of the value of agricultural, forestry, livestock, and fishery services to the area sown to crops, this measures the degree of responsiveness of agricultural socialisation services and characterises the level of socialisation services at the farm household level.

### 3.2.5. Data Sources

In this paper, the implementation of CRFS policy is used as a quasi-natural experiment, and the panel data of 30 provinces (autonomous regions and municipalities directly under the central government) across China (except Tibet, Hong Kong, Macao, and Taiwan) from 2010 to 2021 are selected to analyse the impact of CRFS policy on agricultural carbon emission intensity, taking into account the availability of data. The data on fertilisers; pesticides; agricultural films; agricultural diesel; sown and irrigated areas of crops; total output value of the plantation industry; and total output value of agriculture, forestry, livestock, and fisheries used in this study were obtained from the National Statistical Database (https://data.stats.gov.cn, accessed on 17 July 2017) for all years from 2010 to 2021. The authors manually collated the above data for each province.

## 4. Results

### 4.1. Base Regression Model Regression Results

Based on base regression model, Table 2 reports the regression results of the baseline regression of CRFS on agricultural carbon intensity. Column (1) shows that the estimated coefficients of the core explanatory variable *treat* × *post* are significantly negative when no control variables are added, and its estimated coefficients are always significantly negative at least at the 5 percent level during the gradual addition of control variables, suggesting that the policy of arable land fallow rotations is effective in reducing the regional level of agricultural carbon emission intensity, and Hypothesis H1 is verified. This implies that the implementation of CRFS policy in China has played a positive role in the application of resources. By upgrading the quality of farmland and making up for the shortcomings of agricultural infrastructure, the CRFS policy improves agricultural production efficiency and promotes pesticide and chemical fertiliser reduction as well as the green and low-carbon development of agriculture. The study of Chen et al. (2023) [46] also confirms that the fallow system wheat planting system can effectively improve wheat yield. The fallow crop rotation system reduces the mismatch of agricultural resources, further promotes agricultural technological innovation, improves the fertility of land resources, and reduces the waste of labour resources. For areas with a low fertility of land resources, the fallow crop rotation system promotes the rational cultivation of crops in the area and improves the ecological restoration capacity of the land. As the fallow-tillage rotation system has significantly reduced the intensity of agricultural carbon emissions, China should further promote the steady implementation of the system nationwide and give full play to the positive role of the institutional mechanism in the sustainability of agricultural resources.

**Table 2.** Benchmark regression results of CRFS policy on agricultural carbon emission.

| | (1) | (2) | (3) | (4) | (5) | (6) | (7) |
|---|---|---|---|---|---|---|---|
| | *ACEI* | *ACEI* | *ACEI* | *ACEI* | *ACEI* | *ACEI* | *ACEI* |
| *treat* × *post* | −0.009 *** | −0.008 *** | −0.008 *** | −0.007 *** | −0.006 ** | −0.007 *** | −0.008 *** |
| | (−4.79) | (−3.86) | (−3.74) | (−3.05) | (−2.59) | (−2.86) | (−2.99) |
| *ED* | | 0.025 *** | 0.026 *** | 0.025 *** | 0.024 *** | 0.026 *** | 0.026 *** |
| | | (3.48) | (3.48) | (3.31) | (3.26) | (3.45) | (3.44) |
| *PE* | | | 0.006 | 0.004 | −0.001 | −0.013 | −0.021 |
| | | | (0.17) | (0.11) | (−0.02) | (−0.36) | (−0.56) |
| *SEC* | | | | −0.000 | −0.000 | −0.000 | −0.000 |
| | | | | (−0.51) | (−0.61) | (−0.72) | (−0.82) |
| *BF* | | | | | −0.002 | −0.001 | −0.004 |
| | | | | | (−0.73) | (−0.54) | (−1.19) |
| *SE* | | | | | | 0.739 | 0.842 |
| | | | | | | (1.28) | (1.45) |
| *AOP* | | | | | | | 0.031 |
| | | | | | | | (1.17) |
| *Constant* | 0.122 *** | 0.117 *** | 0.115 *** | 0.119 *** | 0.122 *** | 0.114 *** | 0.103 *** |
| | (107.48) | (68.48) | (9.18) | (8.35) | (8.21) | (7.12) | (5.62) |
| *N* | 360 | 360 | 360 | 360 | 360 | 360 | 360 |
| $R^2$ | 0.065 | 0.098 | 0.099 | 0.099 | 0.101 | 0.105 | 0.109 |
| *Controls* | Yes | Yes | Yes | Yes | Yes | Yes | Yes |
| *id/year* | Yes | Yes | Yes | Yes | Yes | Yes | Yes |

Note: Standard errors of regression coefficients are in parentheses. **, and *** indicate significance levels of 5%, and 1%, respectively.

*4.2. Parallel Trend Test*

In this paper, the event study method was used to regressively analyse the intensity of agricultural carbon emissions for the first six years and the last four years, using 2016 as the baseline. The results of the parallel trend test are shown in Figure 1, when $t < 0$, none of the estimates of $\beta t$ are significant, which indicates that before the implementation of CRFS, there is no significant difference in the intensity of agricultural carbon emissions between the pilot provinces and the non-pilot provinces, but after the implementation of the policy, the estimates of $\beta t$ begin to be significant, which suggests that it is because of the implementation of the policy that the significant difference is caused by the implementation of the policy, and all the coefficients of the confidence intervals of the first six years contain 0, which indicates that the two do not have statistical significance. Therefore, before the implementation of CRFS, there is no significant difference between the experimental and control samples in terms of the intensity of agricultural carbon emissions. Observing the results of the parallel trend test, from the first year after the implementation of CRFS, the coefficient of the effect of CRFS begins to show a decreasing trend, indicating that the effect of CRFS on the intensity of agricultural carbon emissions is significantly negative. Therefore, the parallel trend hypothesis is established.

*4.3. Placebo Test*

In order to ensure that the empirical results on the reduction in carbon emission intensity in agriculture are caused by the implementation of CRFS and to exclude other unknown factors from interfering with the empirical results, this paper further conducts a placebo test. The placebo test is a regression in the sample by randomly selecting any dummy group. In this paper, we conduct a random sample of 500 out of the panel data of 30 provinces and regress it on model (1). Figure 2 reports the results of the placebo test. The distribution of the estimated coefficients unfolds roughly in the centre of 0, and the mean value corresponding to 500 times the estimated coefficients is −0.003, which is very close to 0. It is far away from the true estimated coefficients in this paper, i.e., far away from the vertical line, which implies that the estimated coefficients of the placebo test are

much smaller than the true values, and initially proves the effect of the policy of CRFS on the reduction in the intensity of carbon emissions in agriculture. In addition, in the placebo test, only the estimated coefficient of a 0.2 percent proportion of the sample is larger than the true value of $-0.007$ in the benchmark regression, with a very small probability of being similar to the results of the benchmark regression in this paper, thus providing strong evidence that the implementation of the CRFS policy has an effect on the increase in total factor productivity in agriculture.

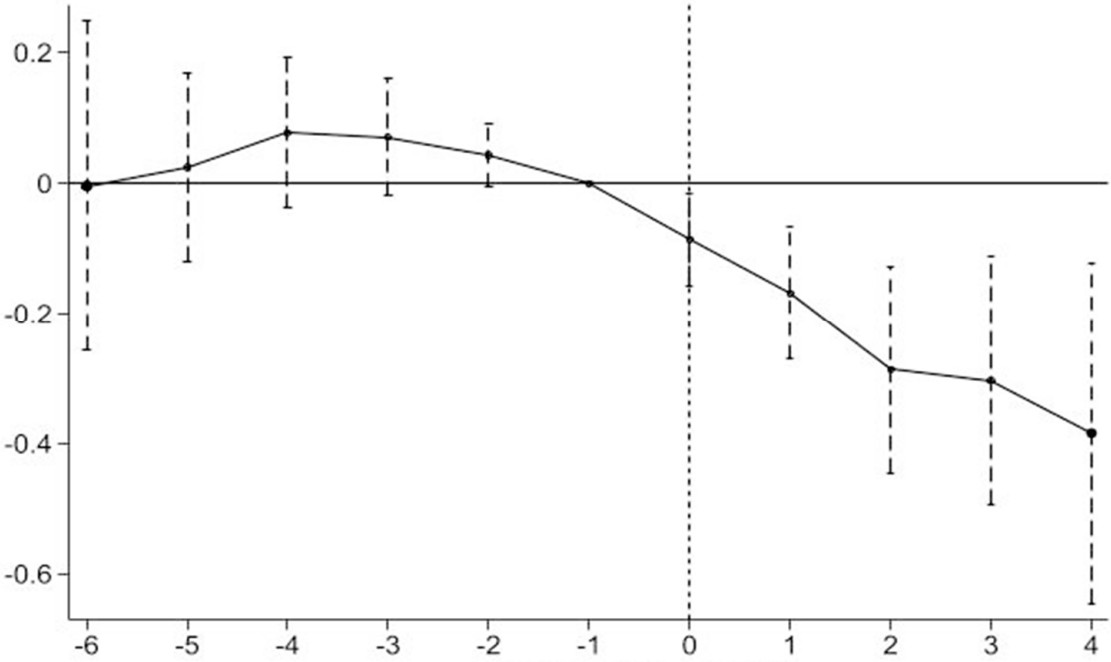

**Figure 1.** Parallel trend test.

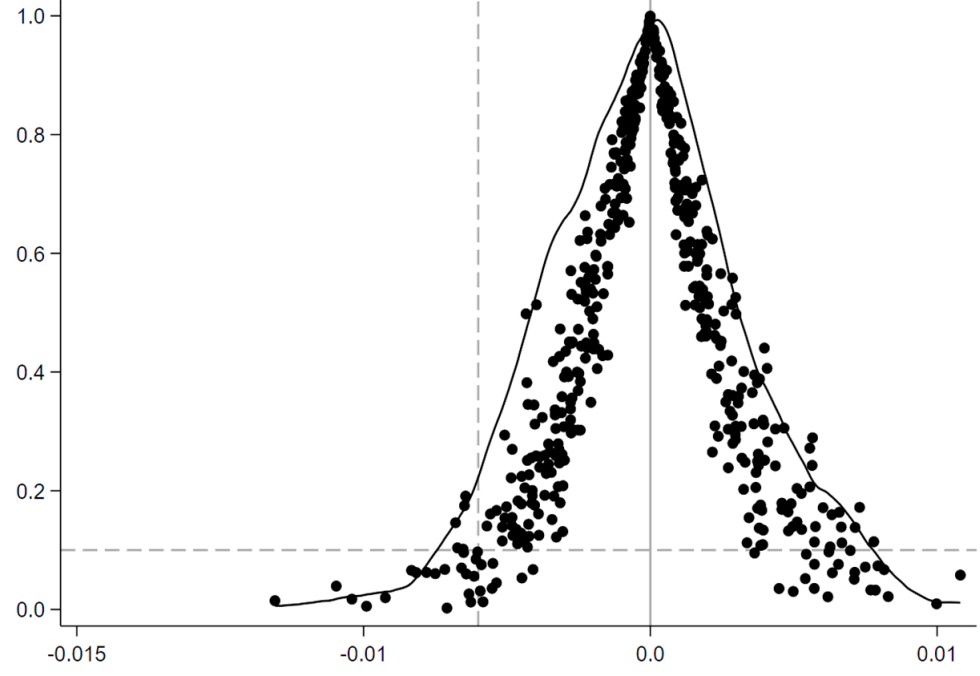

**Figure 2.** Placebo test.

*4.4. Robustness Tests*

4.4.1. Removing Other Policy Distractions

In order to ensure the robustness of the conclusions, a series of robustness tests based on the base regression model are carried out in this paper. Firstly, the replacement of the core explanatory variables is adopted, i.e., excluding other policy interferences for testing. In removing other policy interference, this paper eliminates the impact of the zero-growth policy on fertilisers and pesticides by removing the sample data from 2015 and after; the results show that zero-growth is still significantly negatively correlated at the 1% level, and the results are consistent with the benchmark regression results. Effectively improving the quality of arable land and achieving the sustainable use of arable land are the top priorities of the party and the government, and the construction of high-standard farmland began to be laid out at the central level as early as 1998. In 2011, the State Council formally approved the National Land Rectification Plan (2011–2015). Subsequently, the report of the 20th National Congress of the Communist Party of China (CPC) even explicitly pointed out that all permanent basic farmland should be gradually built into high-standard farmland. By incorporating high-standard construction to eliminate the impact of the high-standard farmland construction policy, the regression results are shown in columns (1)–(2) of Table 3. policies are all significantly negatively correlated at the 5% level, and the results are consistent with the benchmark regression results.

**Table 3.** Robustness results of CRFS policy on agricultural carbon emissions.

| | (1) | (2) | (3) | (4) | (5) | (6) |
|---|---|---|---|---|---|---|
| | *ACEI* | *ACEI* | *ACEI* | *ACEI* | *ACEI* | *ACEI* |
| *treat × post* | | | −0.192 *** | −0.015 * | −0.271 *** | −0.007 *** |
| | | | (−4.78) | (−1.67) | (−4.05) | (−2.63) |
| Zero growth in pesticides and fertilisers | −0.005 ** | | | | | |
| | (−2.03) | | | | | |
| High-standard construction | | −0.007 ** | | | | |
| | | (−2.03) | | | | |
| Constant | 0.113 *** | 0.127 *** | 5.433 *** | 5.721 *** | 5.463 *** | 0.101 *** |
| | (6.24) | (7.90) | (69.89) | (60.02) | (79.71) | (5.13) |
| *Controls* | YES | YES | YES | YES | YES | YES |
| L. *Controls* | NO | NO | NO | NO | NO | YES |
| *N* | 360 | 360 | 360 | 360 | 360 | 330 |
| $R^2$ | 0.096 | 0.096 | 0.911 | 0.423 | 0.714 | 0.081 |
| *id/year* | Yes | Yes | Yes | Yes | Yes | Yes |

Note: Standard errors of regression coefficients are in parentheses. *, **, and *** indicate significance levels of 10%, 5%, and 1%, respectively.

4.4.2. Substitution of Explanatory Variables

In replacing the explanatory variables, this paper uses the agricultural carbon emission level (ACE) to replace the agricultural carbon emission intensity for regression, and the agricultural carbon emission level is measured by the logarithm of agricultural carbon emissions. The empirical results show that the emission reduction coefficient CRFS policy is significantly negatively correlated at the 1% level, and the regression results are shown in column (3) of Table 3, which are consistent with the baseline regression results, verifying the robustness of the conclusions of this paper.

4.4.3. Short-Term Effects

The dates selected for this paper are 2010–2021, so the DID estimates contain the average treatment effect over 11 years, in order to minimise the possibility that a long sample period could make the estimates be influenced by other contemporaneous policies. In view of this, the sample period is re-selected to be 2013–2018, i.e., three years before and two years after the policy implementation. The regression results are shown in column (4) of Table 3, with an abatement coefficient of −0.015 and significance at the 10% level; the

results show that the coefficient of agricultural carbon intensity is significantly negative, consistent with the previous study, which again verifies the robustness of the conclusions.

### 4.4.4. Replacing the Control Variable Setting Method

Differences in resource endowment and economic development between pilot and non-pilot provinces may affect provincial carbon emission intensity to different degrees in trends over time, which may cause estimation bias in the final regression results. In order to control for the effect of differences in characteristics between provinces, an interaction term between the control variables and the time trend is added to the baseline regression so as to exclude the effect caused by the time trend of these characteristic variables. The results obtained are presented in column (5) of Table 3. In addition, in order to mitigate potential bidirectional causality, the control variables in the benchmark regression are borrowed and lagged by one period in their entirety, replacing the original control variables in the benchmark model in order to avoid the endogeneity problem, and the results obtained are shown in column (6) of Table 3. According to the results in the two columns of the table, the estimated coefficients of $treat \times post$ remain significantly negative at the 1 percent level, supporting the results of the benchmark regression.

### 4.4.5. PSM-DID

This paper further uses the propensity score matching method to address the resultant errors caused by selectivity bias, thus ensuring more robust benchmark regression results. The results are displayed in Table 4.The logit model regression is used to predict the propensity score of each province, and then the control variables in model (1) are used as covariates to match the samples using the pairwise 1:4 nearest neighbour matching method (logit), the radius matching method (radius), and the kernel matching method (kernel), to match the control group to the provinces that have implemented the policy of arable land fallow rotation. By matching the control group to the experimental group in this way, the quasi-natural experiment can be made to be closer to random, so as to reduce the endogeneity problem caused by the selection bias of the areas where the "cultivated land fallow and crop rotation" policy is implemented. The estimated coefficients of $treat \times post$ are −0.199, −0.205, and −0.204, respectively, and the significance is consistent with the baseline regression results. At the same time, no matter which matching method is used, the ATT is significantly negative, which indicates that the implementation of the CRFS policy effectively reduces the intensity of agricultural carbon emissions.

**Table 4.** PSM regression results.

|  | (1) | (2) | (3) |
|---|---|---|---|
|  | *logit* | *radius* | *kernel* |
| $treat \times post$ | −0.199 *** | −0.205 *** | −0.204 *** |
|  | (−5.00) | (−5.46) | (−5.55) |
| *Constant* | −1.936 *** | −1.932 *** | −1.890 *** |
|  | (−5.72) | (−5.66) | (−5.59) |
| *N* | 281 | 296 | 318 |
| $R^2$ | 0.960 | 0.960 | 0.960 |
| *Controls* | Yes | Yes | Yes |
| *id/year* | Yes | Yes | Yes |

Note: Standard errors of regression coefficients are in parentheses. *** indicate significance levels of 1%.

### 4.4.6. Quantile Regression

In order to focus on the impact of CRFS policy implementation on the intensity of agricultural carbon emissions at different quantile points, i.e., to accurately explore the policy effect of the implementation of CRFS policy to effectively reduce the intensity of agricultural carbon emissions, a further quantile regression is carried out, and the results of the quantile regression are listed in Table 5 (1)–(3). The following table shows that the

implementation of CRFS policy has a significant reduction effect on agricultural carbon emission intensity, and with the increase in quantile points, the absolute value of the regression coefficient of CRFS policy shows a tendency of increasing and then decreasing (21.1 percent, 21.8 percent, and 15.2 percent, respectively). This suggests that as the year of implementation of the CRFS policy advances, the effect on the two ends of the conditional distribution of agricultural carbon intensity is smaller than the effect on the middle part of it. That is to say, the effect of CRFS policy implementation on low agricultural carbon intensity provinces and high agricultural carbon intensity provinces is relatively small, while the biggest gainers are the intermediate carbon intensity provinces, which indicates that there is a difference in the effect of CRFS policy implementation on the reduction in agricultural carbon intensity.

**Table 5.** Regression results for different quartiles.

|  | (1) | (2) | (3) |
|---|---|---|---|
|  | $q25$ | $q50$ | $q75$ |
| *treat* × *post* | −0.211 *** | −0.218 *** | −0.152 *** |
|  | (−4.78) | (−6.97) | (−4.64) |
| *Constant* | −2.341 *** | −1.756 *** | −1.259 *** |
|  | (−11.55) | (−13.06) | (−5.15) |
| *N* | 360 | 360 | 360 |
| $R^2$ | Yes | Yes | Yes |
| *Controls* | Yes | Yes | Yes |

Note: Standard errors of regression coefficients are in parentheses. *** indicate significance levels of 1%.

*4.5. Examination of the Mechanism of Carbon Emission Reduction in Agriculture*

This paper examines the Mismatch Effect and Pro-Service Effect based on Mechanism Testing Models.

4.5.1. Analysis of the "Mismatch Effect" of the CRFS Policy

"Resource mismatch" is a deviation from the optimal "Pareto efficient allocation". The resource mismatch of agricultural factors of production between different sectors and regions is relatively common, especially labour mismatch and capital mismatch, which inhibits agricultural resource inputs from reaching the optimal level during the agricultural production process, leading to a decrease in desired outputs, such as agricultural output value, agricultural carbon sequestration, and the value of agricultural ecosystem services, and an increase in undesired outputs, such as sources of agricultural environmental pollution, which affects the intensity of agricultural carbon emissions [47]. Based on the previous theoretical analysis, CRFS reduces agricultural carbon emissions by reducing the degree of resource mismatch, and the agricultural carbon emission reduction mechanism test is shown in Table 6. Columns (1)–(2) of Table 6 show that the CRFS policy has a significant negative effect on agricultural resource mismatch, i.e., the CRFS policy will significantly correct agricultural resource mismatch. At the same time, agricultural resource mismatch has a significant negative effect on agricultural carbon emissions, and the mechanism test results confirm that the CRFS policy has a mechanism to inhibit agricultural carbon emissions by correcting agricultural resource mismatch. Therefore, the research hypothesis H2 of this paper is verified.

**Table 6.** Mechanism test results of the impact of CRFS policy on agricultural carbon emissions.

|  | (1) | (2) | (3) | (4) |
|---|---|---|---|---|
|  | *CP* | *ACEI* | *Service* | *ACEI* |
| *treat* × *post* | −0.027 *** |  | 0.170 *** |  |
|  | (−3.63) |  | (4.23) |  |

**Table 6.** *Cont.*

|  | (1) | (2) | (3) | (4) |
|---|---|---|---|---|
|  | *CP* | *ACEI* | *Service* | *ACEI* |
| *CP* |  | −0.883 *** |  |  |
|  |  | (−3.41) |  |  |
| *Service* |  |  |  | 0.011 * |
|  |  |  |  | (1.88) |
| *Constant* | 0.977 *** | 6.901 *** | −0.207 | 0.115 *** |
|  | (7.44) | (17.73) | (−0.65) | (9.04) |
| *N* | 360 | 360 | 360 | 360 |
| $R^2$ | 0.444 | 0.450 | 0.586 | 0.177 |
| *Controls* | Yes | Yes | Yes | Yes |
| *id/year* | Yes | Yes | Yes | Yes |

Note: Standard errors of regression coefficients are in parentheses. * and *** indicate significance levels of 10%, and 1%, respectively.

### 4.5.2. Analysis of the "Pro-Service Effect" of the CRFS Policy

In recent years, the level of socialised agricultural services in China has been steadily increasing, mainly in terms of reforming the production structure of agriculture, forestry, animal husbandry, and fisheries and relying more and more on information service inputs. Agricultural socialisation services can, to a certain extent, make up for the disadvantages of small-scale family management, such as the negative impacts of declining human capital on family agricultural production, based on the theoretical hypothesis of the "agricultural treadmill", and give full play to the effect of "learning by doing" [48], and based on the principle of comparative advantage, through changing the factor structure of agricultural production inputs, they can optimise the level of agricultural socialisation services. Based on the principle of comparative advantage, the synergistic effect of social services and factor structure can be achieved by changing the factor structure of agricultural production inputs and optimising the allocation of agricultural factors, which will in turn affect the level of carbon emissions. Based on the previous theoretical analysis, CRFS reduces agricultural carbon emissions by reducing the degree of resource mismatch, and the agricultural carbon emission reduction mechanism test is shown in Table 7. Columns (3)–(4) of Table 6 indicate that the CRFS policy has a significant positive effect on agricultural socialisation services, i.e., the CRFS policy will significantly promote agricultural socialisation services. At the same time, agricultural socialisation services have a significant positive effect on agricultural carbon emissions, and the results of the mechanism test confirm that the CRFS policy has a mechanism of promoting agricultural carbon emissions through improving agricultural socialisation services. Therefore, the research hypothesis H3 of this paper is verified.

**Table 7.** Heterogeneity analysis results of the impact of CRFS policy on agricultural carbon emissions.

|  | (1) | (2) | (3) | (4) | (5) | (6) | (7) |
|---|---|---|---|---|---|---|---|
|  | **Main Selling Area** | **Balance of Production and Sales Area** | **Main Production Area** | **High Degree of Circulation** | **Low Degree of Circulation** | **Plantation** | **Livestock Area** |
| *treat × post* | −0.030 ** | −0.007 | −0.016 *** | −0.009 ** | 0.003 | −0.020 ** | −0.029 *** |
|  | (−2.31) | (−1.51) | (−4.65) | (−2.20) | (0.41) | (−2.24) | (−4.54) |
| *Constant* | 0.306 *** | 0.051 *** | 0.114 *** | 0.076 *** | 0.030 | 0.264 *** | 0.140 *** |
|  | (14.96) | (3.60) | (6.15) | (3.70) | (1.54) | (9.05) | (9.77) |
| *N* | 84 | 120 | 156 | 180 | 180 | 60 | 300 |
| $R^2$ | 0.582 | 0.755 | 0.300 | 0.149 | 0.487 | 0.838 | 0.567 |
| *Controls* | Yes | Yes | Yes | Yes | Yes | Yes | Yes |
| *id/year* | Yes | Yes | Yes | Yes | Yes | Yes | Yes |

Note: Standard errors of regression coefficients are in parentheses. **, and *** indicate significance levels of 5%, and 1%, respectively.

*4.6. Heterogeneity Analysis*

4.6.1. Heterogeneity in the Positioning of Functional Agricultural Areas

China's three major grain functional zones differ in the positioning of grain production functions, government support, economic development level, etc. In order to verify the difference in the impact of the CRFS policy in different functional zones of grain production, regressions were carried out by distinguishing between the main grain production area, the main marketing area, and the balance of the production and marketing areas. The results show that the carbon reduction effect of the CRFS policy on different food production areas is reflected in the negative and significant effect on the main production areas as well as the main marketing areas at least at the 5% level, but the inhibition effect in the balance of the production and marketing areas is not obvious. The main producing areas, the main reasons for the significant carbon reduction effect of the policy on food production, may be influenced by the following: Firstly, the main food producing areas have assumed more responsibility for food production, and the large amount of investment in agricultural capital has increased the strength of agricultural science and technology innovation; compared with the past, chemical fertilisers, agricultural films, pesticides, and other agricultural consumables have been replaced by cleaner and lower-carbon emerging agricultural science and technology. Under the premise that the total effect remains unchanged, the resulting substitution effect becomes the net effect of carbon reduction. Second, the CRFS policy has boosted the use of agricultural machinery, but the increase in new energy agricultural machinery has led to mechanised production, which was originally a major contributor to carbon emissions from agricultural production, becoming a major contributor to emission reductions, and at the same time boosting the efficiency of agricultural production. The main marketing areas, the main reason for the significant carbon reduction effect of the policy on food production, may be the modernisation of agricultural production in the main marketing areas, which improves the efficiency of pesticides, fertilisers, agricultural machinery, and energy use, coupled with the implementation of the recycling farming model, soil fertilisation, and carbon sequestration, which accelerates the realisation of the carbon emission reduction in agriculture. The reason why the emission reduction effect is not significant in the production and marketing balance area may be that the agricultural land in such functional areas is relatively dispersed, unable to form an effective scale effect of emission reduction, resulting in a relatively low level of agricultural scientific and technological innovation, and the carbon-reducing effect on food production has not yet fully emerged, with a more pronounced latecomer's advantage.

4.6.2. Heterogeneity in the Degree of Land Transfer

In this paper, the level of land transfer is measured by the proportion of the total area of family-contracted arable land transferred to the area of family-contracted operated arable land. Columns (4)–(5) in Table 7 show that the coefficient of the CRFS policy is significantly negative at the 5% statistical level in provinces with a high level of land turnover, indicating that the implementation of the CRFS policy can help to reduce agricultural carbon emissions and promote the development of low-carbon agriculture. This may be due to the fact that CRFS can improve the agricultural cultivation efficiency of farmers, reduce the risk of agricultural business, and promote the popularity of effective agricultural cultivation. At the same time, the implementation of the CRFS policy in provinces with a high degree of land transfer can help to form the effect of the land scale economy, which in turn can help to reduce the intensity of agricultural carbon emissions.

4.6.3. Comparison of Rainfall in Different Years

Given the differences in production between plantation and livestock, this paper divides the plantation and livestock zones for heterogeneity analysis using the 400 mm isoprecipitation line as the boundary. Columns (7) and (8) of Table 7 show that the estimated coefficients for plantation and livestock zones are significant at the 1% and 5% levels, respectively, suggesting that the CRFS policy reduces the intensity of agricultural carbon

emissions in both plantation and livestock zones, and it is more effective in plantation zones than in livestock zones. The possible economic explanation is that the plantation and livestock zones have upgraded their traditional agricultural production facilities through digitalisation, which promotes the precise management of crop growth and livestock rearing and reduces chemical inputs and energy consumption. Intensive management in the livestock area is conducive to promoting the combination of planting and rearing and thus enhancing the efficiency of large-scale operations, promoting the intensive use of pesticides and fertilisers, and enhancing the carbon sequestration capacity of the soil, thus reducing the intensity of agricultural carbon emissions, but as the livestock area is in a semi-arid region with low rainfall, the carbon conversion efficiency is low. This also makes the effect of suppressing carbon emission intensity less effective than that of plantation areas.

### 4.7. Extensibility Analysis

In the context of unbalanced economic development, the CRFS policy provides new opportunities for optimising resource allocation and promoting agricultural social services. From the above, it is clear that the carbon reduction effect of the CRFS policy has begun to take effect. Then, is there a convergence phenomenon in China's regional agricultural carbon emission level? Can the CRFS policy become an accelerator of regional agricultural carbon emission convergence? Answering these questions will help to clarify the important impact of the CRFS policy on regional agricultural carbon emissions and provide a feasible pathway reference for achieving the dual-carbon goal.

The article draws on Li et al. (2023) [49] to establish absolute $\beta$-convergence and conditional $\beta$-convergence models to analyse the regional convergence of CRFS policy and regional agricultural carbon emissions:

$$\ln\left(\frac{ACE_{i,t+1}}{ACE_{i,t}}\right) = \alpha + \beta \ln ACE_{it} + \mu_i + \lambda_i + \delta_{it} \tag{13}$$

where $\ln(ACE_{i,t+1}/ACE_{it})$ is the growth rate of the carbon emission level of province $i$ at time $t$; $\ln ACE_{it}$ is the logarithm of carbon emission level of province $i$ at time $t$; $\mu_i$ is the individual fixed effect; $\lambda_i$ is the time fixed effect; and $\delta_{it}$ is the random perturbation term. $\beta$ is the coefficient of interest in the convergence analysis, and a significant negative value indicates the existence of $\beta$ convergence, while the absolute convergence model assumes that provinces and regions have the same economic characteristics, which is contrary to the reality of the large differences in the development of the region. The conditional convergence model relaxes this assumption, introduces control variables, and can explore the impact of other variables on agricultural carbon emissions, and the conditional convergence model is set as follows:

$$\ln\left(\frac{ACE_{i,t+1}}{ACE_{i,t}}\right) = \alpha + \beta \ln ACE_{it} + \chi CRFS_{it} + \sum_{k=1}^{L} \gamma_k X_{kit} + \mu_i + \lambda_i + \delta_{it} \tag{14}$$

where $X_{kit}$ denotes the $k$th control variable for the provincial domain $i$ at time $t$; $\gamma_k$ is the $k$th control variable coefficient.

The speed of convergence is denoted by $s$, i.e., $\beta$. The larger the absolute value, the faster the speed of convergence.

$$s = -\ln(1+\beta)/T \tag{15}$$

#### 4.7.1. Typical Facts on the Regional Convergence of Agricultural Carbon Emissions in China

Table 8 shows the results of the $\beta$ absolute convergence test using the FE estimation method. It is found that the values of $\beta$ are all significantly negative at least at the 10% level, indicating that there is a significant $\beta$ absolute convergence at the national level, and the growth rate of agricultural carbon emissions in each province is negatively correlated with the initial level, i.e., there is a common convergence trend. The test is carried out by region, and it is found that the values of $\beta$ in the northeast, east, central, and west are all significantly negative, and the absolute value of $\beta$ is the largest in the west, indicating that

the carbon emission reduction level in the west is accelerating to catch up with that of the whole country and also indicating that there is a significant convergence of clubs in the level of agricultural carbon emissions.

**Table 8.** Absolute $\beta$ convergence test results.

|  | (1) | (2) | (3) | (4) | (5) |
|---|---|---|---|---|---|
|  | FE | Northeastern | Eastern Part | Central Section | Western Part |
| $\beta$ | −0.098 *** | −0.163 ** | −0.092 * | −0.088 * | −0.164 *** |
|  | (−4.35) | (−2.80) | (−1.84) | (−1.70) | (−3.04) |
| $\alpha$ | −0.531 *** | −0.977 ** | −0.448 * | −0.527 | −0.878 *** |
|  | (−4.28) | (−2.82) | (−1.74) | (−1.68) | (−3.05) |
| $s$ | 0.103 | 0.178 | 0.097 | 0.092 | 0.179 |
| $N$ | 330 | 33 | 110 | 66 | 121 |
| $R^2$ | 0.569 | 0.901 | 0.408 | 0.822 | 0.593 |
| $id/year$ | Yes | Yes | Yes | Yes | Yes |

Note: Standard errors of regression coefficients are in parentheses. *, **, and *** indicate significance levels of 10%, 5%, and 1%, respectively.

### 4.7.2. Impact of CRFS Policy on Regional Agricultural Carbon Emission Convergence

In order to alleviate the endogeneity problem, the arable land fallow and crop rotation policy is treated with one lag period, and the results are shown in Table 9. When the arable land fallow and crop rotation policy is not taken into account, the convergence coefficient of China's agricultural carbon emission is −0.098, and it is significant at the 1% statistical level, that is, there exists significant conditional β-convergence characteristics of the level of agricultural carbon emissions, the convergence speed of which is 0.103%. Column (2) shows that the conditional β coefficient is −0.026 and is significant at the 5% statistical level, indicating that after the implementation of the policy of arable land fallow and crop rotation, there is still a significant conditional $\beta$ convergence characteristic of the level of agricultural carbon emissions in China, and the speed of convergence at this time is increased from 0.103% to 0.126%. At the same time, the coefficient of the arable land fallow and crop rotation policy is −0.039, which is significant at the 1% statistical level, indicating that the arable land fallow and crop rotation policy can reduce agricultural carbon emissions. After adding control variables in column (3), the coefficient fluctuates less and is more stable, which means that the CRFS policy not only reduces agricultural carbon emissions but also promotes the convergence of its regional carbon emission level. This may be due to the fact that the CRFS policy can reduce the regional emission reduction effect to a certain extent and promote the coordinated development of the region by correcting the degree of mismatch of agricultural resources and promoting agricultural socialisation services.

**Table 9.** Convergence test results for condition $\beta$.

|  | (1) | (2) | (3) |
|---|---|---|---|
|  | ACE | ACE | ACE |
| $\beta$ | −0.098 *** | −0.026 ** | −0.101 *** |
|  | (−4.97) | (−2.48) | (−4.87) |
| $L.CRFS$ |  | −0.039 *** | −0.023 *** |
|  |  | (−8.88) | (−3.83) |
| $\alpha$ | −0.531 *** | −0.145 ** | −0.656 *** |
|  | (−4.90) | (−2.55) | (−5.00) |
| $s$ | 0.103 | 0.126 | 0.106 |
| $N$ | 330 | 330 | 330 |
| $R^2$ | 0.569 | 0.214 | 0.317 |
| Controls | No | No | Yes |
| id/year | Yes | Yes | Yes |

Note: Standard errors of regression coefficients are in parentheses. **, and *** indicate significance levels of 5%, and 1%, respectively.

## 5. Discussion

This study demonstrates the effectiveness of CRFS policy implementation, provides theoretical support for the accurate implementation of related policies and the expansion of the pilot scope of CRFS, and also provides due contribution in the field of agriculture for China to achieve the dual-carbon goal. Firstly, this paper adopts cutting-edge methods in the field of environmental economics to accurately assess the impact of CRFS on ACEI. This provides reliable evidence to reveal the implementation effects of the CRFS. This will not only help to promote the understanding of the system among the agricultural population, reduce resistance, and reduce resistant behaviour but also facilitate the diffusion of the CRFS. Second, analysing the transmission effects of resource mismatch levels and agricultural socialisation services in the implementation of the CRFS in the context of the CRFS will help to understand the intrinsic mechanism of the CRFS on reducing agricultural carbon emissions. Third, the study of the differences in the effects of CRFS policies in the samples of different food functional zones, different degrees of land transfer, and different rainfall areas will help to provide targeted policy recommendations for improving agricultural development policies.

*Limitations and Future Research Directions*

At the same time, this paper also has the following shortcomings: first, there are limitations in the research sample, which is affected by data availability. The research sample in this paper is at the provincial level, which is representative to a certain extent, but it does not go deeper than the municipal level, which affects the universality of the results of the study. Therefore, the research sample can be further expanded or studied in the future. Second, this paper takes agriculture in a narrow sense (i.e., planting) as the research object, and the measurement of agricultural carbon emissions has not penetrated agriculture in a broad sense, so the measurement scope can be further enriched. Considering that the level of agricultural carbon emissions is calculated from the carbon emission coefficients of each of the six types of agricultural carbon sources, including fertilisers, pesticides, agricultural films, agricultural diesel fuel, the area of agricultural land planted with crops and the area of irrigated agricultural land, although it has a certain degree of objectivity, and scientific validity, in the future, we can consider adopting other methods of measurement, such as subdividing the types of carbon sources and establishing a comprehensive system of evaluation indexes for the purpose of measurement.

## 6. Conclusions

At present, global climate change has attracted widespread attention, and reducing carbon emissions has become a common global responsibility and challenge. As a major source of carbon emissions, effectively reducing the intensity of agricultural carbon emissions is of great significance to achieving the dual-carbon goal. CRFS policy is an important means to promote the sustainable development of agriculture and reduce carbon emissions, the study of the impact of cultivated land fallow rotation policy on the intensity of agricultural carbon emissions can provide a strong scientific basis for policy makers, and it is of great significance for enriching the practical experience of reducing the intensity of carbon emissions. In order to demonstrate the validity of the impact of CRFS on ACEI, this paper takes the pilot events of the arable land fallow rotational cropping system in China in 2016, 2017, and 2019 as the research object and verifies the effect of the policy implementation by using a multi-temporal double-difference model.

(1) The results of the benchmark regression show that CRFS can significantly reduce the intensity of agricultural carbon emissions, and the results remain robust after the results are verified by multiple methods.

(2) The results of mechanism analysis show that correcting factor mismatch and promoting agricultural socialised services are important pathways for the arable land fallow rotational cropping system to reduce the intensity of agricultural carbon emissions.

(3) The results of heterogeneity regression show that the CRFS is more conducive to promoting the reduction in agricultural carbon emission intensity in the main grain production area, main grain marketing area, high land transfer area, and plantation area than in the grain production and marketing balance area, low land transfer area, and animal husbandry area.

## 7. Policy Recommendations

The above findings confirm the agricultural carbon emission reduction effect of the CRFS policy. Therefore, this paper puts forward the following policy recommendations. First, governments at all levels should strengthen the CRFS and give full play to its ecological effect in carbon emission reduction. CRFS increases the publicity of the CRFS policy, which is believed to reduce people's income and thus affect the promotion of the system. In this regard, the government should publicise the advantages of the system through village committees and village cadres to promote a deeper understanding of the effects of the policy. The government should also promote multi-objective co-construction and optimise the implementation of the policy in order to enhance the carbon emission reduction effect of the policy by promoting arable land fallow rotations according to local conditions.

Secondly, governments at all levels should pay great attention to correcting the mismatch of agricultural resources and promoting the role of agricultural social services in the carbon reduction process of the CRFS policy. Therefore, first of all, they should reasonably allocate agricultural scientific and technological labour resources, accelerate the flow of labour factors between regions and industries, and optimise the allocation of agricultural resources. The rational allocation of various agricultural resources makes the original physical and chemical properties of arable land and the structure of biological populations improved, thus enhancing the sustainable use of land power. Secondly, the government should increase its support for socialised service projects, cultivate the main body of agricultural socialised service projects, and strengthen the socialised service capacity of grassroots villages and dragon-head enterprises. By promoting agricultural socialised services through a top-down model, agricultural carbon emissions will be reduced, thus accelerating the realisation of the dual-carbon goal.

Thirdly, governments at all levels should differentiate the implementation of CRFS policies according to local conditions and precision. The carbon reduction effect of the CRFS policy varies significantly among different land transfer levels and agricultural functional positioning zones. For provinces with a low degree of land transfer, land transfer should be accelerated to strengthen the inhibiting effect of CRFS on agricultural carbon emissions. Each functional agricultural zone should adjust its agricultural structure and production methods, promote high agricultural quality, and balance the relationship between food production capacity and the carrying capacity of resources and the environment, with a view of achieving agricultural modernisation.

**Author Contributions:** X.Z.: conceptualisation, formal analysis, methodology, writing—original draft, software. H.S.: funding acquisition. X.X.: supervision, project administration, review and editing, Z.Y.: supervision, project administration, review and editing, S.Z.: data curation, validation. All authors have read and agreed to the published version of the manuscript.

**Funding:** This research was supported by the National Natural Science Foundation of China (grant number: 71963030).

**Data Availability Statement:** All data generated or analysed during this study are included in this article in the form of figures and tables. Additional information about the dataset or the dataset in a different format than what is presented in this article can be obtained from the corresponding author upon request.

**Conflicts of Interest:** The authors declare that they have no known competing financial interests or personal relationships that could have appeared to influence the work reported in this paper.

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
