# Peer review of "Can a Crop Rotation and Fallow System Reduce the Carbon Emission Intensity of Agriculture?"

_land, doi:10.3390/land13030293_

Round 1
Reviewer 1 Report
Comments and Suggestions for Authors
Dear authors,
The research you have done is very important because this topic is relevant worldwide. The work is deep but has several inconsistencies.
There is a lot of talk about abandoning fallow, and you choose it as one of the objects of study. I would like a deeper literary justification of why the fallow is suitable for agriculture and for what reasons it is to be criticized.
In the structure of the introduction, it would be good to state the aim as short as possible (one sentence is recommended). The aim should be at the end of the introduction.
A research hypothesis should also be one sentence long, and yours are cover more than a page.
The methods section should clearly state each method used along with the formulas. In your case, it's hard to tell where the methods are and where the results are.
The results section should present the identified results after applying certain methods.
Several conclusions should be formulated, because part of the information presented in the conclusions should be placed in the introduction section.
Conclusions and discussion should be separate because it is difficult for the reader to understand the main clearly defined results. I miss this very much in this article. Lines 755 to 766 should be part of the introduction, and 777 to 794 should be part of the discussion of the results but not the conclusions.
I suggest reviewing the requirements for submitting a literature review.
The article should be heavily revised, considered and resubmitted.
Reviewer 2 Report
Comments and Suggestions for Authors
1.Why are questions used in the title?
2.The article writing format needs further improvement, with more writing problems. For example, the corresponding author is not indicated and line 314 is incorrectly written.
3.Images in the paper are too blurry, improve clarity and enlarge the images.
4.Data sources are not labeled to be listed in the references.
5.Does the raw data need to be listed, or listed as an attachment.
6.Conclusion and discussion are separated, and the conclusion is placed at the end of the text.
Comments on the Quality of English LanguageLanguage quality needs to be further improved
Reviewer 3 Report
Comments and Suggestions for Authors
Dear Authors,
Thank you for your interest in publishing with the Land journal. Unfortunately, at this time, I am unable to suggest your manuscript for publication. The article lacks a complete discussion section!!!! I encourage you to fully develop the discussion and resubmit the article.
Additionally, please make the following revisions:
-
Adhere to the author's instructions as outlined here: https://www.mdpi.com/journal/land/instructions
-
Please condense the abstract to a maximum of 200 words.
-
For keywords, avoid using words that are already in the title of the article.
-
Ensure consistency in your citations throughout the text.
Regarding the content: You mention that "traditional cultivation requires large quantities of pesticides, fertilizers, and chemicals such as plastic films." Could you please clarify which specific chemical films are being referred to here?
Comments on the Quality of English Language
Minor editing.
Round 2
Reviewer 1 Report
Comments and Suggestions for Authors
Dear authors,
I am glad of your improved article, but still some things are correctable.
Firstly, the aim should be in the end of introduction. The reader could dismiss trying to find where is the main research aim. The aim should be just one sentence.
Secondly, how you are using references in the text. You should check recommendations for authors, because "superscript" is not applied.
and thirdly, I recommend to change the chapters name. Chapter 4 "Results and discussion" should be changed to "Results". The chapter "Discussion" should be after all results chapter. Conclusion and recommendations should be separated in to different chapter.
Fix it and good luck
